# Genome-Wide Identification and Expression Analysis of the Trehalose-6-phosphate Synthase and Trehalose-6-phosphate Phosphatase Gene Families in Rose (*Rosa hybrida* cv *‘Carola’*) under Different Light Conditions

**DOI:** 10.3390/plants13010114

**Published:** 2023-12-31

**Authors:** Yingdong Fan, Peng Gao, Tong Zhou, Siyu Pang, Jinzhu Zhang, Tao Yang, Wuhua Zhang, Jie Dong, Daidi Che

**Affiliations:** 1College of Horticulture and Landscape Architecture, Northeast Agricultural University, Harbin 150030, China; fanying_dong@163.com (Y.F.); gaopeng@neau.edu.cn (P.G.);; 2Key Laboratory of Cold Region Landscape Plants and Applications, Harbin 150030, China

**Keywords:** *Rosa hybrida*, trehalose-6-phosphate synthase gene, trehalose-6-phosphate phosphatase gene, trehalose, photoperiod and light intensity responses, flowering

## Abstract

Trehalose, trehalose-6-phosphate synthase (TPS),and trehalose-6-phosphatase (TPP) have been reported to play important roles in plant abiotic stress and growth development. However, their functions in the flowering process of *Rosa hybrida* have not been characterized. In this study we found that, under a short photoperiod or weak light intensity, the content of trehalose in the shoot apical meristem of *Rosa hybrida* cv ‘Carola’ significantly decreased, leading to delayed flowering time. A total of nine *RhTPSs* and seven *RhTPPs* genes were identified in the genome. Cis-element analysis suggested that *RhTPS* and *RhTPP* genes were involved in plant hormones and environmental stress responses. Transcriptome data analysis reveals significant differences in the expression levels of *RhTPSs* and *RhTPPs* family genes in different tissues and indicates that *RhTPPF* and *RhTPPJ* are potential key genes involved in rose flower bud development under different light environments. The results of quantitative real-time reverse transcription (qRT-PCR) further indicate that under short photoperiod and weak light intensity all *RhTPP* members were significantly down-regulated. Additionally, *RhTPS1a*, *RhTPS10*, and *RhTPS11* were up-regulated under a short photoperiod and showed a negative correlation with flowering time and trehalose content decrease. Under weak light intensity, *RhTPS11* was up-regulated and negatively regulated flowering, while *RhTPS5*, *RhTPS6*, *RhTPS7b*, *RhTPS9*, and *RhTPS10* were down-regulated and positively regulated flowering. This work lays the foundation for revealing the functions of *RhTPS* and *RhTPP* gene families in the regulation of rose trehalose.

## 1. Introduction

Light is a paramount environmental factor that profoundly influences the flowering time of plants at three distinct levels: photoperiod, light quality, and light intensity [1]. Moreover, light facilitates photosynthesis and serves as an indispensable energy source for plants as it results in sugar production [2], which plays a significant role in the process of flower bud differentiation [3].Trehalose is a non-reducing disaccharide composed of two glucose molecules, widely distributed in living organisms, and its metabolic precursor is trehalose-6-phosphate [4]. Plant trehalose serves as a crucial growth-regulating substance within plants, exhibiting distinct biological activities and participating in the regulation of various physiological processes [5]. It aids plants in coping with environmental stresses such as high temperature [6], freezing [7], and drought [8]. Trehalose-6-phosphate primarily functions as a mediator of sugar signaling during the onset of flowering, playing a significant role in the regulation of seed germination, leaf growth, development and flowering time, as well as plant senescence [9]. Plants synthesize trehalose through a conserved, two-step metabolic pathway. The first step involves the conversion of glucose from UDP-glucose to glucose-6-phosphate by trehalose-6-phosphate synthase (TPS), resulting in the production of trehalose-6-phosphate (T6P). The T6P is then dephosphorylated into trehalose by trehalose-6-phosphate phosphatase (TPP).

The *TPS* and *TPP* genes exist since the early stages of green plant lineage formation. Eleven *TPS* genes were encoded by the Arabidopsis and rice genomes, while ten and twelve *TPP* genes were encoded by the Arabidopsis and rice genomes, respectively [10]. Phylogenetic analysis showed that *AtTPS* gene was divided into two different branches: Class I (AtTPS1-AtTPS4) and Class II (AtTPS5-AtTPS11) [11]. Class I AtTPS members have a conserved N-terminal TPS-like domain and a less conserved C-terminal TPP domain. Each AtTPS member in Class II is also a dual-domain protein with an N-terminal TPS-like domain and a highly conserved C-terminal TPP domain [12]. TPP proteins in all plants are distinguished by the presence of a conserved phosphatase domain, and all AtTPPs encode functional TPP enzymes. Additionally, although TPPs share similar activities, their patterns of differential expression suggest that they may have functions related to specific tissues, stages, or processes [13].

The current knowledge on the number and structural characteristics of *TPS* and *TPP* members in other plant species is limited, and the functions of these genes remain unclear. The relationship between TPS (trehalose-6-phosphate synthase) and TPP (trehalose-6-phosphate phosphatase) proteins is closely linked to flowering time in plants. In Arabidopsis, plants that have been genetically modified to overexpress *AtTPS1* exhibit smaller leaves, earlier flowering, and a denser growth pattern than wide type WT [14]. In contrast, overexpression of *AtTPPA* led to larger leaves, later flowering, and reduced branching compared to WT [14]. The *AtTPS1a* gene silenced in *A. thaliana* resulting in 25–30% decrease of Tre6P content, delayed flowering and an increase in the number of rosette leaves [15]. In shoot apical meristems, the *AtTPS1* interacts with the age pathway by suppressing miR156 expression which leads to the accumulation of SQUAMOSA promoter-binding protein-like 3 (SPL3), SPL4, and SPL5 to promote plant flowering [15]. By mutating *AtTPPI* in Arabidopsis, researchers observed a delay in flowering. When the *AtTPPI* gene was introduced into the *tppi* mutant, the gene restored the normal flowering phenotype of Arabidopsis [16]. Additionally, the *RAMOSA3* (*RA3*) gene encoding a TPP protein in maize, determined the inflorescence structure and potentially increasing yield [17].

*Rosa hybrida* is currently the largest cut flower in terms of global planting area and sales volume (https://aiph.org/, accessed on 9 February 2023) [18]. The *TPS* and *TPP* gene families have been reported to be of great significance in many kinds of plants’ response to stress and regulation of growth and development [19,20]. With the update of gene structure annotation technology (https://gitee.com/CJchen/IGV-sRNA, accessed on 9 February 2023), rose *TPS* genes need to be reidentified and analyzed. Furthermore, there have been no studies to evaluate the evolution and expression profiles of the *TPP* gene family in *Rosa hybrida*. In this study, the member of *TPS* and *TPP* family were identified in roses, and their structures, phylogenetic relationships, and cis-acting elements were thoroughly analyzed. The expression patterns of *TPS* and *TPP* in different tissues of roses were analyzed using transcriptome data. Additionally, the crucial role of *RhTPS* and *RhTPP* in flower formation in response to various photoperiods and light intensities were investigated by transcriptome data and qRT-PCR. Finally, we conducted a correlation analysis between flowering time, trehalose content, and *RhTPSs/TPPs* gene expression to identify key functional genes. The findings from this study serve as a valuable biological foundation for gaining a deeper understanding of the functionalities of the *RhTPS* and *RhTPP* gene families in rose.

## 2. Results

### 2.1. The Flowering Time and Trehalose Content of the R. hybrida cv ‘Carola’ Were Influenced by Both Photoperiod and Light Intensity

Under different light treatments, the flowering time of *R. hybrida* cv ‘Carola’ was influenced. As shown in Figure 1a,b, under the CK treatment, roses flowered around 45 days after treatment. However, when the photoperiod was changed to 10 hL/14 hD or 8 hL/16 hD, the flowering time of roses was extended to 55 and 60 days, respectively, indicating a significant delay in flowering. When the light intensity was reduced to 80% and 60% of CK, roses failed to flower (Figure 1a,b). Additionally, the floral initiation rate of roses also varied under different light treatments. As the duration of light intensity decreased, the floral initiation rate of roses decreased from 90% to 75% and then to 60%. Roses under 80% and 60% light intensity treatments completely lost the ability to initiate flowers and became sterile (Figure 1c).

To further investigate the reasons for delayed flowering and decreased floral initiation rate, the trehalose content of the apical meristem of roses under different light treatments was measured. Under the CK treatment, the trehalose content was approximately 0.25 mg·g^−1^FW (Figure 1d). As the photoperiod changed to 10 hL/14 hD and 8 hL/16 hD, the trehalose content decreased to 0.13 mg·g^−1^FW and 0.12 mg·g^−1^FW (Figure 1d), representing a decrease of 48% and 52%. As shown in Figure 1e, when the light intensity was reduced to 80%, the trehalose content decreased by 88% from 0.25 mg·g^−1^FW to 0.02945 mg·g^−1^FW. For the light intensity treatment at 60% LI, the trehalose content was 0.023381 mg·g^−1^FW, which was 90.64% lower than that of the CK group.

### 2.2. Identification, Analysis, and Characteristics of TPS/TPP Family Members in Rosa hybrida

Nine *RhTPS* and seven *RhTPP* genes were identified from the rose genome and named respectively as *RhTPS1a* to *RhTPS11*, *RhTPPA* to *RhTPPJ* according to the orthologs in *A. thaliana*. The results indicate that the coding sequence (CDS) length of *RhTPSs*/*RhTPPs* is 2565–2934/360–1173 bp (Appendix A). The full length of nine RhTPS was predicted to be 854–977 amino acids (Appendix A). The instability index of RhTPS ranged from 45.88 to 50.36, with an average of −0.391 to −0.164. On the other hand, the RhTPP family encoded shorter proteins, ranging from 119 to 390 amino acids. The instability index of RhTPP was 28.54 to 49.24, with an average gravity of −0.555 to −0.27 (Appendix A). Additionally, the theoretical isoelectric point indicated that the RhTPSs were acidic proteins (pI 5.71–6.23), while RhTPPs were classified as basic proteins (pI 5.19–9.16) (Appendix A). Both RhTPS and RhTPP were hydrophilic. Predication of subcellular location showed that all RhTPS and RhTPP proteins localized to the cytoplasm or mitochondrial (Appendix A) in a manner of related to their function.

### 2.3. Evolutionary Relationship of TPS and TPP in Different Species

A phylogenetic tree was constructed by maximum likelihood method illustrating the evolutionary history of TPS and TPP protein among *R. hybrida*, *R. rugosa*, *R. wichruana*, *F. vesca*, *M. domestica*, *P. trichocarpa,* and *A. thaliana* (Figure 2). These TPSs were divided into class I (Group I) and class II (Group II–V). Class II TPS proteins were assigned to four major groups (Group II–V) with high bootstrap support. In particular, the nine RhTPS proteins often clustered closer to the FvTPS proteins, compared with the arrangements in other species (Figure 2a). In addition, RhTPS1a and RhTPS1b clustered close to AtTPS1 in the Group I (Figure 2a). Overall, the TPS proteins in the Rosaceae family exhibit a relatively high level of conservation, displaying a consistent evolutionary trend.

A maximum likelihood (ML) phylogenetic tree was also constructed using the full-length protein sequences of 7 RhTPPs, 9 AtTPPs, 7 FvTPPs, 11 MdTPPs, 10 PtTPPs, 6 RrTPPs, and 7 RwTPPs, and showed that these TPP proteins were classified into 4 groups (Figure 2b). Moreover, within each group, the distribution of the seven RhTPPs was asymmetrical, with two members in Group I, two in Group II, two in Group III, and one in Group IV. Overall, the RhTPPs exhibited closer phylogenetic relationships with those of *F. vesca* and *M. domestica*, as opposed to those of *A. thaliana*.

### 2.4. Domain, Gene Structure Analysis and Motif

Based on evolutionary analysis, RhTPS is divided into three categories named TPSI, TPSII, and TPSIII, while RhTPP is also divided into three categories named TPPI, TPPII, and TPPIII (Figure 3a,b). All RhTPS proteins contained trehalose OtsA and trehalose-phosphatase domains (Figure 3a). In contrast, the RhTPP proteins contained only trehalose-phosphatase domains (Figure 3d). Compared to TPSII and TPSII, genes in TPSI have a higher number of introns (Figure 3b). Furthermore, a total of 10 conserved motifs were found in *RhTPS* and *RhTPP* gene families (Figure 3c). The specific amino acid sequences of motifs of RhTPSs and RhTPPs and the detailed information of the MEME sites analysis were shown in the Appendix A, respectively. Most RhTPSs exhibit relatively consistent motif composition with 10 conserved motifs, RhTPS1a and RhTPS1b contain 6 motifs (Figure 3c). In comparison with RhTPS, RhTPP had fewer motifs. All RhTPPs had motifs 1 and 6 (Figure 3), whereas motif 10 was found only in RhTPPA and RhTPPD2 (Figure 3), possible because of its long evolutionary history. RhTPPF has the least number of motifs, which consist of motif 1, 6 and 8 (Figure 3). These different organizations of conserved motifs might correlate to the different function of these RhTPSs and RhTPPs.

### 2.5. Chromosomal Locations and Gene Duplication of the RhTPS and RhTPP Genes in R. hybrida

Chromosome distribution analysis found that these 16 *RhTPSs*/*TPPs* were unevenly distributed on 6 out of 7 *R. hybrida* chromosomes (Figure 4a), of which chr7 contained 4 members, chr2, chr4 and chr5 harbored 3 members, chr1 had 2 members, chr1 had 1 member and chr6 did not have any members (Figure 4a). The results of the collinearity analysis showed that there were four segmental duplication events which occurred in the gene pairs *RhTPS9*–*RhTPS10*, *RhTPS7a*–*RhTPS7b*, *RhTPPA*–*RhTPPF*, *RhTPPD1*–*RhTPPJ* (Figure 4b). Given that the majority of the Ka/Ks values were found to be less than one (Appendix A) it can be inferred that strong purifying selection pressure and limited functional divergence have acted on the *RhTPS* and *RhTPP* gene family following tandem duplication.

### 2.6. Putative Cis-Acting Regulatory Elements in the Promoter Region of RhTPS and RhTPP Genes

Cis-acting elements of analysis of the promoters of *RhTPS* and *RhTPP* unveiled that a significant portion of the identified motifs exhibited responsiveness to both environmental stress and phytohormones. Four types of hormone-responsive regulatory elements were also found, including auxin-responsive elements (AuxRR-core, TGA-element, TGA-box) in three *RhTPSs* and four *RhTPPs*, gibberellin-responsive elements (P-box, GARE-motif and TATC-box) in five *RhTPSs* and five *RhTPPs*, MeJA-responsiveness (TGAGC-motif and CGTCA-motif) in five *RhTPSs* and seven *RhTPPs*, ABA responsive elements (ABRE) in all *RhTPSs* and six *RhTPPs* (Figure 5). Moreover, light-related elements and the MYB transcription factor binding elements were present in all 16 *RhTPS* and *RhTPP* genes (Figure 5), suggesting that *RhTPS* and *RhTPP* genes may be interacting with MYB transcription factors to regulate plant development and growth.

### 2.7. Expression Profiles of the RhTPS/TPP Genes in Various Organs and Developmental Stage

Based on the publicly available transcriptome data, the expression profiling of 16 *RhTPS/TPP* genes in different rose tissues and responding in various light treatment were further investigated. Compared with other *RhTPS* genes, *RhTPS1b*, *RhTPS7b*, *RhTPS6*, *RhTPS10*, and *RhTPS5* had higher transcript levels in most tissues (Figure 6a). In addition, *RhTPS6* was highly expressed in roots and *RhTPPA* was highly expressed in leaves and stamens. *RhTPS3* and *RhTPPH* were expressed at very low levels in all tissues (Figure 6a). The results showed that 14 *RhTPS*/*TPP* genes were expressed in shoot apical meristem development at different light treatment, except for *RhTPS7a* and *RhTPS3*, which had particularly low expression. Among them, *RhTPS6*, *RhTPS10*, *RhTPS5*, *RhTPPA,* and *RhTPS1b* were expressed in higher level (Figure 6b). Additionally, compared with normal conditions, both *RhTPPF* and *RhTPPJ* showed a significant decrease at 8 h (short light length) and 60% light intensity (low light intensity).

### 2.8. Expression Pattern of RhTPSs and RhTPPs in Response to Light Signal and Regulation of Flowering in Rose

With the reduction of photoperiod or light intensity, the flowering time of *Rosa hybrida* cv ‘Carola’ increased, and the floral initiation rate decreased (Figure 1a–c). Significant changes in the expression levels of some members of the *RhTPS*/*TPP* family also occurred. We analyzed the *RhTPS* and *RhTPP* genes expression patterns of *Rosa hybrida* in response to light signal using qRT-PCR. The expression of *RhTPS1a*, *RhTPS1b*, *RhTPS10*, and *RhTPS11* showed a significant increase after the photoperiod was reduced from 12 h to 10 h and 8 h (Figure 6c). Among them, the expression patterns of *RhTPS1a* and *RhTPS10* gradually changed with the decrease of the photoperiod (Figure 6c). However, the expression of *RhTPS1b* significantly increased when the photoperiod changed to 8hL/16hD, and the expression of *RhTPS11* was lower in the CK group and enhanced after the decrease of photoperiod (Figure 6c). *RhTPS5*, *RhTPS6*, *RhTPS7a*, *RhTPS7b*, and *RhTPS9* showed the strongest expression when the photoperiod was 10 hL/14 hD (Figure 6c). They exhibited weaker expression in the CK group and the 8 hL/16 hD group (Figure 6c). The expression patterns of members of the *RhTPP* family were completely different from those of the RhTPS family. All members of the *RhTPP* family were highly expressed in the CK group (Figure 6c). Once the photoperiod was reduced, their expression decreased, reaching the lowest level in the 8 hL/16 hD group (Figure 6c).

The expression patterns of members of the *RhTPS* and *RhTPP* families under different light intensities were different. *RhTPS1a*, *RhTPS5*, *RhTPS6*, *RhTPS7a*, and *RhTPS7b* showed enhanced expression in the CK group (Figure 6d). On the other hand, the expression of *RhTPS1b*, *RhTPS9*, and *RhTPS11* was strongest when the light intensity was 80% (Figure 6d). Unlike the above genes, *RhTPS10* showed increased expression under a light intensity of 60%, while its expression decreased in the 80% light intensity and CK groups (Figure 6d). All members of the *RhTPP* family were highly expressed in the CK group (Figure 6d). However, reducing the light intensity to 80% or 60% greatly suppressed the expression of *RhTPP* family members (Figure 6d).

### 2.9. Correlations between RhTPS/TPP Gene Expression Levels, Flowering, and Trehalose Content

The correlation analysis was conducted between the flowering time, flower-forming rate, apical meristem trehalose content, and the expression levels of *RhTPS* and *RhTPP* genes in roses treated under different photoperiods and light intensities. As shown in Figure 6e, all members of the *RhTPP* family were negatively correlated with the flowering time, indicating that the earlier the flowering time, the higher the expression level of *RhTPPs*. Conversely, *RhTPS1b* expression was positively correlated with the flowering time (Figure 6e) and as *RhTPS1b* expression increased, the flowering time was delayed. The rose flower-forming rate was positively correlated with the trehalose content under different photoperiod treatments (Figure 6e), indicating that trehalose played a positive role in rose flower formation. The expression levels of *RhTPPs* were positively correlated with the trehalose content, while the expressions of *RhTPS10*, *RhTPS1a*, and *RhTPS11* were negatively correlated with the trehalose content. The expressions of *RhTPPJ*, *RhTPPD2*, *RhTPPH*, and *RhTPPF* were positively correlated with the flower-forming rate.

Under different light intensity treatments, *RhTPS9* expression was negatively correlated with flowering time (Figure 6f) and the higher the expression of *RhTPS9*, the earlier the flowering time. Conversely, *RhTPS7a* expression was positively correlated with flowering time (Figure 6f). The expression of *RhTPPs*, *RhTPS9*, *RhTPS5*, *RhTPS6*, *RhTPS7b*, and *RhTPS10* were positively correlated with trehalose content (Figure 6f). Additionally, the flower-forming rate of roses under different light intensities was highly positively correlated with trehalose content and the expression of *RhTPS9*, *RhTPS5*, *RhTPS10*, and *RhTPPs* (Figure 6f). However, *RhTPS11* expression was negatively correlated with flower-forming rate and trehalose content (Figure 6f).

## 3. Discussion

### 3.1. Role of Trehalose Metabolism in Rose Formation in Response to Different Light Environment

Shorter photoperiods or lower light intensities can cause poor flower bud development or delayed flowering, which affects the quality of ornamental plants and severely limits crop yields. Unlike previous research which suggested that CF (also known as recurrent, perpetual, everbearing, or remontant flowering) roses are day-neutral plants [21], this study found that *R. hybrida* cv ‘Carola’ is sensitive to different light environments. The flowering time and rate of *R. hybrida* cv ‘Carola’ were influenced by both the photoperiod and the intensity of light (Figure 1a–c). *R. hybrida* cv ‘Carola’ treated with short photoperiods and low light intensity had significantly lower trehalose content in their shoot apical meristem compared to the CK group (Figure 1d,e). Research has also shown that the majority of cultivated plants have low levels of trehalose [22], which is consistent with the results of this study. Trehalose is widely present in almost all living organisms and plays a crucial role in both in vitro and in vivo conditions [23]. It is also considered to be a key organic osmotic regulatory substance effectively involved in plant abiotic stress tolerance [24]. In many varieties of lotus flowers such as ‘Boli Furen’ and ‘Xue Lian14′, flower bud abortion occurs under weak light conditions, accompanied by a decrease in trehalose content by 89% [25]. Additionally, the interaction between T6P and SnRK1 is closely related to stress-induced seed abortion during early maize grain development [26]. In response to weak light environments, the reduction of carbohydrates (especially trehalose) in aborted flower buds is one of the key factors that lead to delayed flowering time and reduced flowering rate [25]. Ultimately, this results in flower buds dying from reduced energy supply and affected sugar signaling, which suggests that an appropriate amount of trehalose plays an important role in normal flowering processes in plants.

### 3.2. Characterization and Evolution of RhTPS and RhTPP Genes in Rose

In this study, a total of nine *RhTPSs* and seven *RhTPPs* were identified (Appendix A). A multiple species phylogenetic tree classified members of the TPS and TPP gene families from seven species into five groups and four groups, respectively (Figure 2). The classification is different from that in poplar and wheat [27,28], but similar to that in rice and sugarcane [29,30]. *RhTPS1a* and *RhTPS1b* have more exons than other *RhTPS* family members, indicating that *RhTPS* genes may have experienced different selective pressures during evolution, resulting in exon loss and the large number of differences in exon number between *RhTPSs* genes. While the majority of the RhTPP structural domain was located towards the C-terminus, the RhTPS structural domain was clustered at the N-terminal region or within the central section of the protein. This observation is consistent with reports for Arabidopsis, rice, and cotton [31,32], highlighting that the characteristic domain structure of this protein family is conserved across different species. The removal of the N-terminal extension region immediately preceding the TPS domain has been shown to increase TPS activity compared to the full-length protein [33]. TPS enzymes are characterized by their high catalytic potential, and the N-terminal region serves as an inhibitory domain that regulates TPS activity [10]. In the *RhTPP* family, only co-linearity between *RhTPPA*, *RhTPPF*, *RhTPPD1* and *RhTPPJ* was identified. However, there are 10 *TPP* genes (*AtTPPPA*-*AtTPPJ*) in Arabidopsis which have undergone multiple genome duplication events [10]. Eight out of the ten genes are paralogous pairs. The reduction of such duplication events suggests that roses have undergone different evolutionary patterns compared to Arabidopsis.

### 3.3. Analysis of Cis-Acting Elements in the Promoter of RhTPS and RhTPP Genes

Transcriptional activation levels are coordinated by upstream cis-acting elements which are crucial for plant responses to environmental conditions. The *RhTPS* and *RhTPP* promoters contain some common motifs and numerous repetitive regions. All *RhTPS* and *RhTPP* promoters contain MYB elements (Figure 5) which are important for plant development and stress response [34,35]. Inducing OsMYB30 through low-temperature signaling in rice, which activates the expression of *OsTPP1* gene, leads to the accumulation of fucose in seeds and α-Inhibition of amylase activity, thereby inhibiting seed germination at low temperatures [36]. Furthermore, the widespread light-responsive cis-acting elements on the *RhTPS* and *RhTPP* promoters suggested that the expression of these genes might be related to plant responses to different light conditions, implying a potential connection between the Tre6P pathway and light signaling. This study found some ABA signal response elements on the *RhTPS* and *RhTPP* promoter regions (Figure 5), suggesting the enormous potential of *RhTPS* and *RhTPP* in transforming plants and creating drought-resistant and stress-resistant plants. *OsTPS8* may regulate suberin deposition in rice through ABA signaling, and overexpression of *OsTPS8* is sufficient to confer enhanced salinity tolerance without any yield penalty [37]. ABA can induce *AtTPPE* expression, and *TPPE* mutants are insensitive to ABA, while plants that overexpress *AtTPPE* are hypersensitive to ABA [38]. The ABA metabolism pathway is one of the important pathways for plants to resist abiotic stress. Abiotic stress changes cell osmotic pressure, leading to the accumulation of ABA as a stress response [39].

### 3.4. Expression Patterns of RhTPSs and RhTPPs under Different Light Treatment

We further investigated the expression patterns of *RhTPS* and *RhTPP* genes under different light treatments. As shown in Figure 6c,d, many *RhTPP* genes and some *RhTPS* genes may be related to various aspects of low light stress. In addition, most of these genes may respond to both short photoperiod and low light intensity treatments. For example, *RhTPS9*,*10* and *RhTPPs* (Figure 6c,d). In Arabidopsis, *TPS9* has the function of regulating flower transformation in leaf vascular tissues. Long day light cycle stimulation leads to increased chromatin accessibility and increased expression of *TPS9* [40]. In roses, reducing light exposure time appropriately (10 h of light) and decreasing light intensity (80% of light intensity) can promote the expression of *RhTPS9*. Therefore, the mode of action of TPS9 may be inconsistent in roses and Arabidopsis. We speculate that *RhTPS9*,*10* and *RhTPPs* may play a crucial role in the flowering of roses in response to different light environments. The expression of *RhTPPs* in response to light environment is relatively consistent (Figure 6d), indicating that *RhTPPs* may have consistent functions. In contrast, all 13 *MdTPS* genes in apples responded to exogenous sucrose application, indicating that *TPS* may play a consistent role in apple response to sucrose treatment [41]. This also reflects the differential expression of *TPS* and *TPP* in different species.

### 3.5. The Relationship between Light, Trehalose, Flower Formation, RhTPSs and RhTPPs

In *R. hybrida* cv ‘Carola’, reducing photoperiod activates the expression of most *RhTPS* members such as *RhTPS1a*, *RhTPS1b*, *RhTPS10*, and *RhTPS11*, but suppresses the expression of all *RhTPP* members (Figure 6c). After reducing light intensity, the expression of most *RhTPS* and *RhTPP* family members is suppressed (Figure 6d). Regardless of reducing the photoperiod or decreasing light intensity, the ultimate result is a decrease in trehalose content and a delay in flowering time and flowering rate. It has been reported that T6P (an intermediate product in the *TPS*-T6P-*TPP* pathway) in plants can regulate the expression of *FT* to promote the key flowering genes *SOC1*, *LFY*, *AP1*, and so on by responding to sugar signals [15]. Moreover, Arabidopsis plants lacking *AtTPS1* will experience a delayed transition from vegetative growth to flowering [42]. In lotus, *TPS1*- overexpression lotus showed significantly decreased flower bud abortion rates both in normal-light and low-light environments [25]. The overexpression of *RhTPS* genes in roses to improve flowering rate or flowering time still needs further experimental validation. However, the results of this experiment preliminarily suggest that *RhTPS1b* is a negative factor, which is inconsistent with previous studies. This study found that members of the *TPP* family seem to have more important functions, and all members of the *RhTPPs* showed a positive correlation with flowering (Figure 6c,d). *RhTPPJ*, *RhTPPD2*, *RhTPPH*, and *RhTPPF* play positive roles in rose response to different photoperiods during flowering process. In other plants, members of the *TPP* family have been widely reported to participate in plant stress response and yield improvement. For example, overexpression of *OsTPP1* in rice promotes seed germination and increases rice yield and stress resistance [43]. In wheat, *TaTPP7* promotes seed grain filling [44]. However, whether the changes of *RhTPS*/*TPP* genes in roses are caused by stress environments or due to different metabolic patterns compared to other plants still needs to be further studied through experiments.

The influence of light intensity on the flowering of *R hybrida* cv ‘Carola’ appears to be even greater, as the flowering rate drops to 0% when light intensity is reduced (Figure 1c). Through correlation analysis, this experiment has identified *RhTPS9*, *RhTPS5*, and *RhTPS10* as key genes promoting the flowering of roses, and they have also been identified as key genes involved in trehalose synthesis. In potatoes, *TPS9* has been reported to participate in the positive regulation of potato cold resistance [45]. *AtTPS5* positively regulates resistance to gray mold but negatively regulates resistance to pseudomonas syringae [46]. This is consistent with our finding that *RhTPS9*/*5* also participate in the stress response of roses and play key roles. When stress was applied to the roses in this experiment, the functions of *RhTPS* and *RhTPP* changed. The functions of *RhTPS* and *RhTPP* under non-stressed conditions still need to be further validated using transgenic technology.

Based on our findings and previous research evidence, we propose a model in which *RhTPS*/*TPP* regulate the content of trehalose to participate in the response of roses to different photoperiods and light intensity during flowering (Figure 7). This may provide valuable clues for further analysis of the specific functions of *RhTPS*/*TPP* genes in roses.

## 4. Materials and Methods

### 4.1. Plant Materials and Treatments

In this study, *Rosa hybrida* cv ‘Carola’ was obtained from the Key Laboratory of Cold Region Landscape and Application, Northeast Agricultural University, Harbin, China (45.6811° N,126.6250° E). Using cuttage for propagation expansion, each plant was pruned 0.5 cm above the third leaflet from the base of the shoot. The plants were then grown in plant incubators under five different conditions. All treatments are shown in Table 1. Samples were taken from the apical meristems of five treatment groups, with one group being sampled as soon as flowering commenced in each treatment. At least three biological replicates were used to collect all sample points. The samples were rapidly frozen using liquid nitrogen and then stored at −80 °C.

### 4.2. Determination of Trehalose Content in R. hybrida cv ‘Carola’

The high-performance liquid chromatography (HPLC) method was employed to analyze the trehalose content [47]. A sample of 0.3 g was ground into a powder form for 25 s at a frequency of 60 Hz. Subsequently, powdered sample was transferred into a 10 mL tube pre-filled with 4 mL of 80% ethanol. The tube was then subjected to extraction in a water bath at 37 °C for 60 min, with intermittent mixing. After extraction, the tube was centrifuged at 10,000 rpm for 10 min at a temperature of 4 °C. The supernatant was cautiously transferred to a fresh 10 mL centrifuge tube. The process was repeated once more with 4 mL of 80% ethanol. The resulting supernatant was combined and thoroughly mixed, and the total volume was adjusted to 10 mL. Next, 1 mL of the extract was transferred to a 2 mL centrifuge tube and subjected to freeze-centrifugation for drying. Subsequently, 1 mL of ultrapure water was added to the dried sample, followed by rotary evaporation. The sample was shaken until fully dissolved and then centrifuged at 10,000 rpm for 10 min at 4 °C. The supernatant was then filtered through a 0.22 μm aqueous membrane, after which 200 uL was deposited into a 1.5 mL HPLC injection vial with pre-lined tubes and set aside. The sugar separation method was conducted following the manufacturer’s instructions with slight modifications. The Agilent HPLC column (ZORBAX Carbohydrate column with 4.6 × 150 mm, 5 μm; Agilent, Santa Clara, CA, USA) was utilized to detect the content of trehalose.

### 4.3. Identification of RhTPS and RhTPP Genes in R. hybrida and Collection of Their Physicochemical Properties

To screen the TPS/TPP proteins in rose, the Hidden Markov Model (HMM) profile of Glyco_transf_20 domain (PF00982) and Trehalose PPase domain (PF02358) were used for local search based on the whole-genome protein sequences (https://www.rosaceae.org/analysis/282, accessed on 11 February 2023) [18,48] by HMMER3.0 with the E-value ≤ 1 × 10^−10^ as the threshold. Furthermore, 11 AtTPSs and 10 AtTPPs protein sequences were used to perform a BLASTP search against the local protein database with the threshold of E-value < 1 × 10^−5^. The potential *R. hybrida* TPS/TPP proteins were obtained by integrating the results of the HMMER (http://www.hmmer.org/, accessed on 12 February 2023) and BLASTP searches. Furthermore, all candidate protein sequences were further confirmed by SMART (http://smart.embl-heidelberg.de/, accessed on 12 February 2023) [49], and InterPro (http://www.ebi.ac.uk/interpro/, accessed on 13 February 2023) [50]. The candidates were submitted to the ExPASy (http://web.expasy.org/protparam/, accessed on 13 February 2023) [51] database to investigate the physicochemical properties such as molecular weight (Mw), isoelectric point (pI), instability index (II), aliphatic index (AI), and grand average of hydrophobicity (GRAVY). The subcellular localization of them was predicted by the CELLO tool (v2.5) (http://cello.life.nctu.edu.tw/, accessed on 14 February 2023) [52].

### 4.4. Phylogenetic Relationships, Gene Structure, Conserved Motif and Cis-Element Analysis

The TPS and TPP phylogenetic relationships of *R. hybrida*, *A. thaliana*, *Populus trichocarpa*, *Malus domestica*, *Fragaria vesca*, *R. Rogusa* and *R. wichurana* were reconstructed by MEGA11 [53] with the muscle program to perform multiple sequence alignments and maximum likelihood method (ML) analyses, with the bootstrap value set at 1000 repetitions. A gene structure map of the members of the *RhTPS* and *RhTPP* gene families was generated using the R packages ggbio and GenomicRanges. The conserved protein motifs were predicted using online MEME tools (http://alternate.meme-suite.olrg/suite.olrg/, accessed on 20 February 2023) with the following parameters: a maximum of 10 motifs, allowing any number of repetitions, and an optimal width ranging from 6 to 250. The 2000 bp sequence upstream of the coding region was selected to predict the cis-acting elements of *RhTPS* and *RhTPP* genes. This prediction was conducted using the PlantCARE software (https://bioinformatics.psb.ugent.be/webtools/plantcare/html/, accessed on 25 February 2023). To visualize the enrichment of each cis-acting element in the *RhTPS* and *RhTPP* promoters, the TBtools software [54] was utilized.

### 4.5. Gene Chromosomal Location and Collinearity Analysis

The position of each *RhTPS* and *RhTPP* genes on seven chromosomes was obtained from the GDR (https://www.rosaceae.org/, accessed on 1 March 2023) and visualized with TBtools. MCscan X software [55] was used to analyze the gene replication events and identify the genes with segmental duplication and tandem repeats. Visualization made use of TBtools.

### 4.6. Expression Profile Analysis of RhTPSs and RhTPPs Based on RNA-seq Data

The previously published transcriptomic data (PRJNA725601) was used to investigate the flower bud differentiation in *R. hybrida* under various photoperiods and light intensities (photoperiods: 12 hL/12 hD, 10 hL/14 hD, 8 hL/16 hD; light intensities: 100%LI, 80%LI, 60%LI) at day zero and day three, respectively. Tissue-specific expressions of *RhTPS* and *RhTPP* gene family members in different tissues of the *R. hybrida* were downloaded from the NCBI Sequence Reading Archive database (PRJNA546486). The HISAT2 (v2.1.0) [56] and StringTie (v1.3.5) pipeline [57] were used to calculate the value of fragments per kilobase per million (FPKM). The expression profiles of *RhTPSs* and *RhTPPs* were retrieved and visualized with heat maps by the superheat package in R software.

### 4.7. Quantitative Real-Time PCR (qRT-PCR)

The expression level of *RhTPS/TPP* genes of *R. hybrida* cv ‘Carola’ were detected in the shoot apical meristematic tissues. Total RNA was extracted by FastPure Universal Plant Total RNA Isolation Kit (Vazyme Biotech Co., Ltd., Nanjing, China), and synthesis cDNA by HiScript II QRT SuperMix for qPCR (Vazyme Biotech Co., Ltd., Nanjing, China) following the manufacturer’s protocols. The *RhACTIN2* was used as reference gene [21]. The relative expression of *RhTPSs*/*TPPs* was quantified using the ChamQ Universal SYBR qPCR Master Mix (Vazyme Biotech Co., Ltd., Nanjing, China) and BIO-RAD CFX96 Touch system (CFX Touch; Bio-Rad, Hercules, CA, USA). The qRT-PCR thermal cycling condition is as follows: 95 °C for 30 s, followed by 40 cycles of 10 s at 95 °C, and 30 s at 60 °C. The 2^−∆∆^CT quantification method was used for calculating the relative expression level [58]. Each biological replicate was repeated three times with three technical repetitions. The primers used in the current study are listed in Appendix A.

### 4.8. Data Analysis

Data were analyzed using GraphPad Prism 9.0 (GraphPad Software, Boston, MA, USA, www.graphpad.com, accessed on 5 March 2023) in order to compare the differences between treatments. The figures display the mean ± standard deviation (SD) of biological triplicates unless otherwise indicated. One-way analysis of variance was employed to determine statistical significance.

Correlation analysis was adopted using Pearson’s correlation. The package ggcorrplot was used for visualizations, and *p*-values less than 0.05 (*p* < 0.05) were displayed.

## 5. Conclusions

This study demonstrated that a short photoperiod or low light intensity treatments significantly decreased the trehalose content in the shoot apical meristem of the *R. hybrida* cv ‘Carola’, leading to delayed flowering time and decreased flower formation rate. It indicates that trehalose plays an important role in the normal flowering of roses. Nine *RhTPSs* and seven *RhTPPs* were identified, and the phylogenetic trees of TPS and TPP homologous proteins from seven plants (including *R. hybrida* and six other plants) were constructed, which were divided into five groups and four groups, respectively. Most cis-acting elements in the promoter regions of *RhTPS* and *RhTPP* members were related to plant hormones, especially the ABA hormone. Under short-day or low light intensity conditions, the expression levels of all *RhTPP* family members decreased significantly. *RhTPS1a*, *RhTPS10*, and *RhTPS11* were identified as the key inhibitors of rose flowering under short-day treatment, while *RhTPPs* promoted flowering. Under weak light intensity treatment, RhTPS11 was the key inhibitor of rose flowering, and *RhTPPs*, *RhTPS5*, *6*, *7b*, *9*, and *10* may positively regulate the flowering process. This study provides a foundation for further understanding of the function of trehalose in the flowering of roses.

## Figures and Tables

**Figure 1 plants-13-00114-f001:**
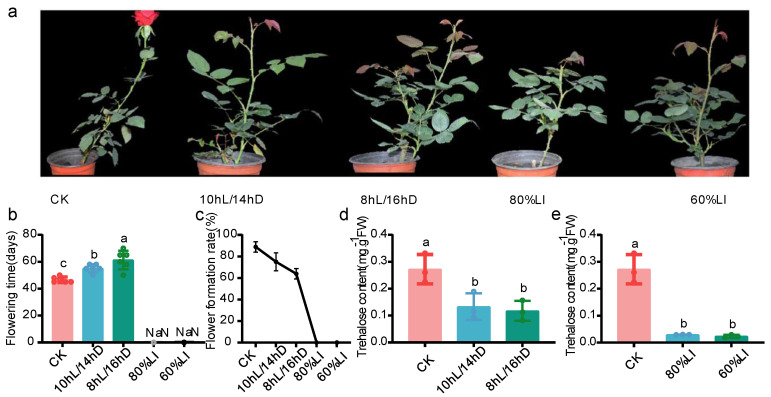
Flowering time and trehalose content data under different lighting treatments. (**a**) Phenotypic characterization of *R. hybrida* cv ‘Carola’ under CK, 10 hL/14 hD, 8 hL/16 hD, 80%LI, 60%LI condition. (**b**) Flowering time of *R. hybrida* ‘Carola’ under CK, 10 hL/14 hD, 8 hL/16 hD, 80%LI, 60%LI condition. Values are mean ± SEM (*n* = 6). (**c**) Flower formation rate of *R. hybrida* cv ‘Carola’ under CK, 10 hL/14 hD, 8 hL/16 hD, 80%LI, 60%LI condition. Values are mean ± SEM (*n* = 6). (**d**) Trehalose content (mg·g^−1^FW) of *R. hybrida* cv ‘Carola’ under CK, 10 hL/14 hD, 8 hL/16 hD condition, values are mean ± SEM (*n* = 3). (**e**) Trehalose content (mg·g^−1^FW) of *R. hybrida* cv ‘Carola’ under CK, 80%LI, 60%LI condition, values are mean ± SEM (*n* = 3). Different letters indicate significant differences.

**Figure 2 plants-13-00114-f002:**
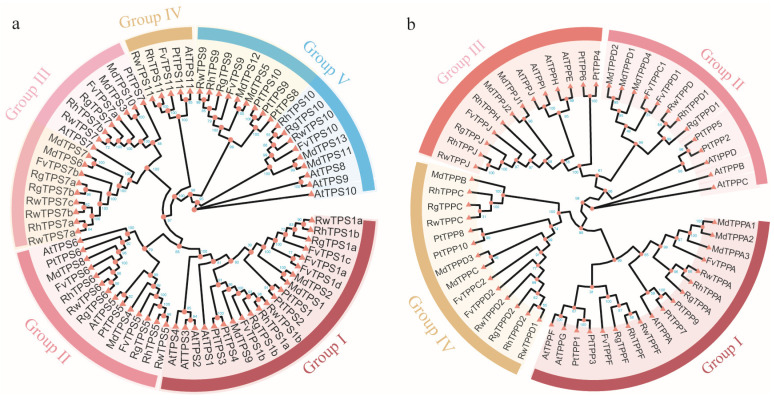
Construction of phylogenetic tree proteins for TPS and TPP. (**a**) Phylogenetic tree of the TPS proteins from seven plant species. The TPS proteins of *R. hybrida* (RhTPS), *P. trichocarpa* (PtTPS), *A. thaliana* (AtTPS), *Malus domestica* (MdTPS), *Fragaria vesca* (FvTPS), *Rosa rugosa* (RrTPS) and *Rosa wichruana* (RwTPS). (**b**) Phylogenetic tree of the TPP proteins from seven plant species. The TPP proteins of *R. hybrida* (RhTPP), *P. trichocarpa* (PtTPP), *A. thaliana* (AtTPP), *M. domestica* (MdTPP), *F. vesca* (FvTPP), *R. rugosa* (RrTPP) and *R. wichruana* (RwTPP).

**Figure 3 plants-13-00114-f003:**
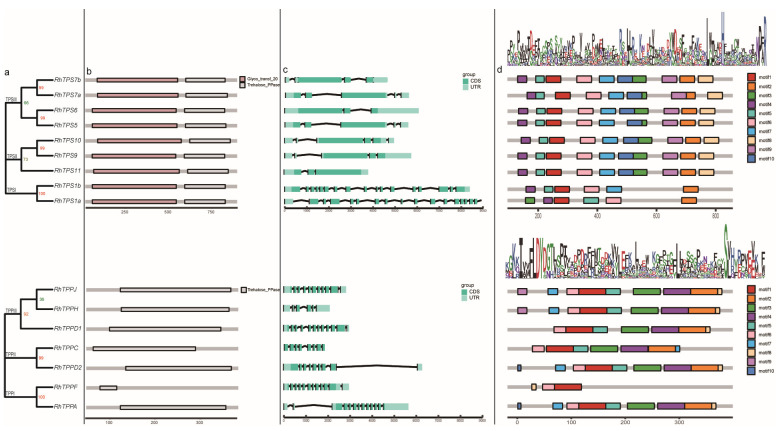
Phylogenetic relationships and gene structures, domains, and motifs of the RhTPSs and RhTPPs family. (**a**) Phylogenetic analysis. (**b**) The conserved trehalose-6-phosphate synthase (TPS) domain (Glyco_transf_20), and the trehalose-6-phosphate phosphatase (TPP) domain (Trehalose_PPase)—the domains are indicated in different color. (**c**) Light green rectangles represent untranslated regions (UTRs); turquoise rectangles represent coding sequence (CDS) or introns; black lines represent introns. (**d**) Motif analysis, all motifs were identified by MEME tools, as shown in different bars.

**Figure 4 plants-13-00114-f004:**
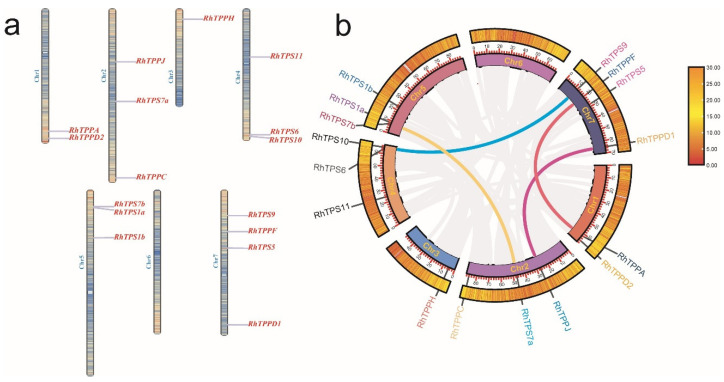
Chromosomal synteny analysis of *RhTPS* and *RhTPP* genes. (**a**) Chromosome distribution and positioning of RhTPSs and RhTPPs across all seven chromosomes of rose. (**b**) Synteny analysis of *RhTPS* and *RhTPP* genes of *R. hybrida*. Chr1, Chr2, Chr3, Chr4, Chr5, Chr6, and Chr7 represent the seven chromosomes of *R*. *hybrida*, respectively.

**Figure 5 plants-13-00114-f005:**
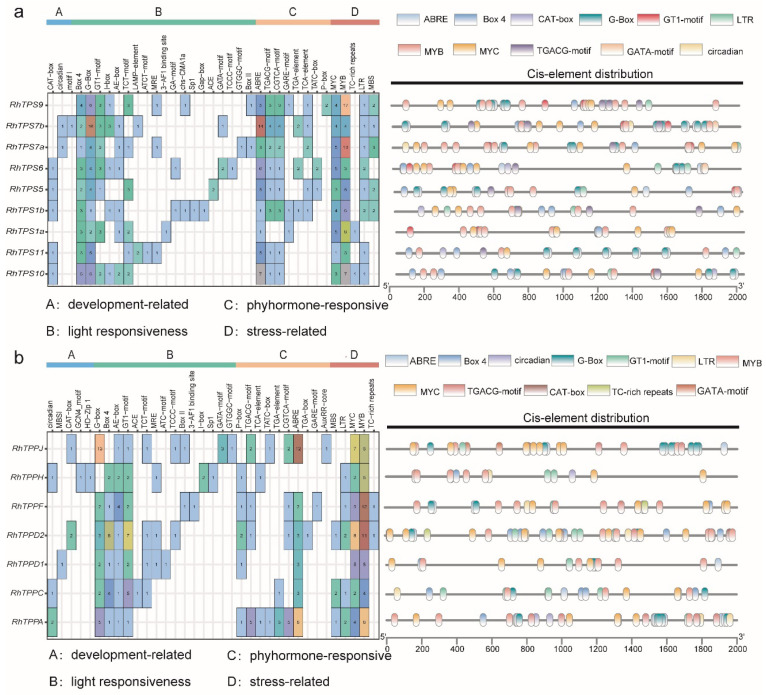
Composition and number of cis-acting elements in the promoter regions of *RhTPS* (**a**) and *RhTPP* (**b**) genes. (**a**) Cis-acting elements found in the promoter region of *RhTPSs*. (Left): The number and function classification of cis-acting element in each *RhTPS* genes. (Right): Key cis-acting elements in each *RhTPS*; elements are represented by the boxes in different colors. (**b**) Cis-acting elements found in the promoter region of *RhTPPs*. (Left): The number and function classification of cis-acting element in each *RhTPP* genes. (Right): Key cis-acting elements in each *RhTPP*; elements are represented by the boxes in different colors.

**Figure 6 plants-13-00114-f006:**
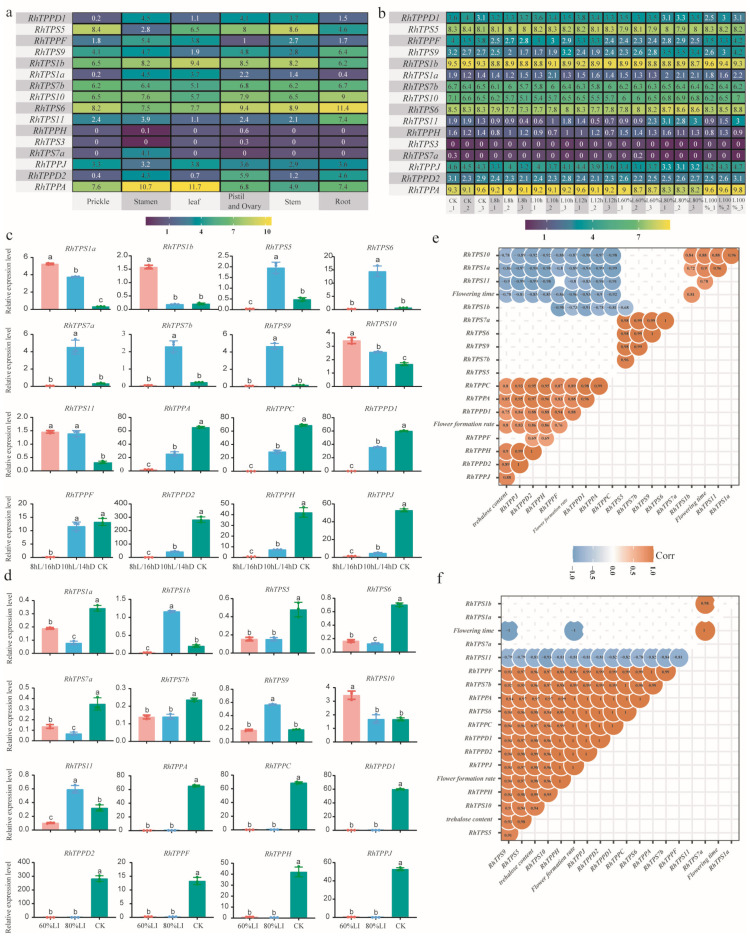
Expression profiles of *RhTPS* and *RhTPP* genes in different tissues and under diverse light environments. (**a**) Expression profile of *RhTPSs* and *RhTPPs* in different tissues. FPKM values were normalized by log_2_(FPKM + 1) transformation to display the heatmap color scores. (**b**) Expression profiles of *RhTPSs* and *RhTPPs* after three days under different light treatments. Different shades of yellow and purple denote the extent of the expression values according to the colour bar provided log_2_(FPKM + 1). (**c**) Expression levels of *RhTPS* and *RhTPP* genes under CK, 10 hL/14 hD, 8 hL/16 hD. *RhACTIN2* was used as the reference gene. Error bars indicate SD (*n* = 3). (**d**) Expression levels of *RhTPS* and *RhTPP* genes under CK, 80%LI, 60%LI. *RhACTIN2* was used as the reference gene. Error bars indicate SD (*n* = 3). (**e**) Correlation analysis of *RhTPS* and *RhTPP* gene expression with flowering time, flowering rate, and trehalose content under CK, 10 hL/14 hD, 8 hL/16 hD conditions. Use the Pearson correlation coefficient method for correlation analysis, where orange represents positive correlation and dark blue represents negative correlation. The correlation coefficient is kept within the circle. Values with *p* < 0.05 are retained. (**f**) Correlation analysis of *RhTPS* and *RhTPP* gene expression with flowering time, flowering rate, and trehalose content under CK, 80%LI, 60%LI conditions. Use the Pearson correlation coefficient method for correlation analysis, where orange represents positive correlation and dark blue represents negative correlation. The correlation coefficient is kept within the circle. Values with *p* < 0.05 are retained.

**Figure 7 plants-13-00114-f007:**
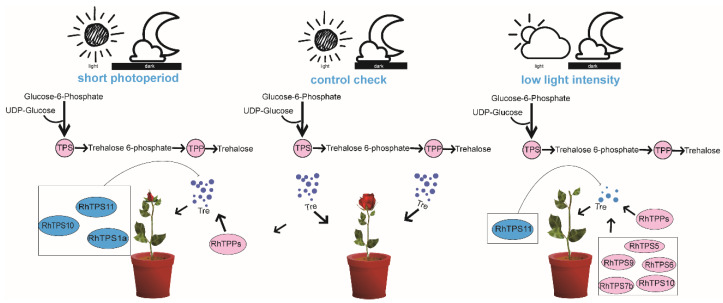
Proposed *RhTPPs* and *RhTPPs* genes respond to short light periods and weak light intensity through different expression patterns. A solid arrow indicates activation, and a line ending in a bar indicates negative adjustment.

**Table 1 plants-13-00114-t001:** Treatment of different photoperiods and light intensities.

Photoperiod	Light Intensity	Temperature	Humidity	Hereafter
12 h light and 12 h dark	200 μmol·m^–2^·s^–1^	25 °C	40%	CK
10 h light and 14 h dark	200 μmol·m^–2^·s^–1^			10 hL/14 hD
8 h light and 16 h dark	200 μmol·m^–2^·s^–1^			8 hL/16 hD
12 h light and 12 h dark	160 μmol·m^–2^·s^–1^			80%LI
12 h light and 12 h dark	120 μmol·m^–2^·s^–1^			60%LI

## Data Availability

The data presented in this study are available upon request from the corresponding author.

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
