# Peer review of "Genome-Wide Identification and Expression Analysis of the Trehalose-6-phosphate Synthase and Trehalose-6-phosphate Phosphatase Gene Families in Rose (Rosa hybrida cv ‘Carola’) under Different Light Conditions"

_plants, 2023, doi:10.3390/plants13010114_

Round 1

Reviewer 1 Report

Comments and Suggestions for Authors

I checked your manuscript and described comments below.

This paper is very good reserach about the genome-wide gene expression analysis of trehalose-6-phosphate synthase (TPS) and trehalose-6-phosphate phos-3 phatase (TPP) in Rose (Rosa hybrida) by changing the light conditions.

I think you should consider the following points.

1.       This study used the old software MAGA X for Phylogenetic analysis. I think it's better to use the latest version of MEGA 11.

2.       This paper applies stress using different light conditions, but there is a similar paper that studies differences in gene expression due to heat stress.

Xiao-Ru Wei et al., Horticulturae, 2022, 8, 429. https://doi.org/10.3390/horticulturae8050429

3.       I think it would be better to emphasize the novelty of this paper a little more.

I don't think this paper has new various major mistakes or grammatical problems.

Author Response

For research article

Response to Reviewer 1 Comments

1. Summary

Thank you very much for your comments and professional advice. Those comments are all valuable and very helpful for revising and improving our paper, as well as the important guiding significance to our researches. We have studied comments carefully and have made correction which we hope meet with approval. For clarity, we will use red text to show our revisions.

2. Point-by-point response to Comments and Suggestions for Authors

Comments 1:

This study used the old software MEGA X for Phylogenetic analysis. I think it's better to use the latest version of MEGA 11.

Response 1:

Thank you very much for your attention and suggestions on our research. After team discussion, we fully agree with your suggestion and will use MEGA 11 for phylogenetic analysis in the revised version. In actual use, we also found that the running speed of MEGA11 has been improved compared to MEGAX. The specific changes are shown below, and we have also redrawn the evolutionary tree. We hope to receive your approval.

4.4. Phylogenetic relationships, gene structure, conserved motif and cis-Element analysis

The TPS and TPP phylogenetic relationships of R. hybrida, A. thaliana, Populus trichocarpa, Malus domestica, Fragaria vesca, R. rogusa and R. wichurana were reconstructed by MEGA11 [55] with the muscle program to perform multiple sequence alignments and Maximum Likelihood Method (ML) analyses, with the bootstrap value set at 1000 repetitions.

55. Tamura, K.; Stecher, G.; Kumar, S. MEGA11: Molecular Evolutionary Genetics Analysis Version 11. Molecular biology and evolution 2021, 38, 3022-3027, doi:10.1093/molbev/msab120.

Figure 2. Construction of phylogenetic tree proteins for TPS and TPP.

(a) Phylogenetic tree of the TPS proteins from seven plant species. The TPS pro-teins of R. hybrida (RhTPS), P. trichocarpa (PtTPS), A. thaliana (AtTPS), Malus domestica (MdTPS), Fragaria vesca (FvTPS), Rosa rugosa (RrTPS) and Rosa wichruana (RwTPS).

(b) Phylogenetic tree of the TPP proteins from seven plant species. The TPP proteins of R. hybrida (RhTPP), P. trichocarpa (PtTPP), A. thaliana (AtTPP), M. domestica (MdTPP), F. vesca (FvTPP), R. rugosa (RrTPP) and R. wichruana (RwTPP).

Comments 2:

  This paper applies stress using different light conditions, but there is a similar paper that studies differences in gene expression due to heat stress. Xiao-Ru Wei et al., Horticulturae, 2022, 8, 429. https://doi.org/10.3390/horticulturae8050429

Response 2:

We are very grateful for the reviewer's inquiries. Wei et al. have made significant contributions to the identification and analysis of the rose TPS gene family, particularly in studying the impact of different heat stress conditions on TPS gene expression. However, we simultaneously identified and analyzed two gene families involved in trehalose synthesis, TPS and TPP, rather than focusing solely on one TPS gene family. Because plants synthesize trehalose through a conserved, two-step metabolic pathway. The first step involves the conversion of glucose from UDP-glucose to glucose 6-phosphate by trehalose-6-phosphate synthase (TPS), resulting in the production of treha-lose-6-phosphate (T6P). The T6P is then dephosphorylated into trehalose by trehalose 6-phosphate phosphatase (TPP). Furthermore, trehalose has been extensively reported to participate in plant responses to both high and low temperatures [19]. Especially, our research has uniqueness in addressing the differential gene expression under light conditions.

Specifically, there are several differences between our article and Wei et al.'s article:

Firstly, our study not only identified one gene family, but also simultaneously identified members of two gene families and studied their expression patterns. More and more studies are simultaneously focusing on the expression patterns of two gene families in response to stress in plants [27]. Our study is the first to simultaneously identify and analyze the genes of TPS and TPP family members in roses, which can better characterize the mode of action of the fucose metabolism pathway in roses and provide a more detailed basis for future research. This comprehensive study can provide more comprehensive gene family information, helping us better understand the functions and interactions of trehalose related gene families in organisms.

Secondly, the methods and logic we analyzed differ significantly from Wei's paper. One reason is that the scientific issues we focus on are inconsistent. We focus on the flowering phenotype of roses, and during the sampling process, we take stem tip meristems closely related to flowering. Wei et al. focused on the heat tolerance of rose cuttings, so the sampling time span was shorter than our study. And we provided the phenotypic changes of the processed roses in our paper, which were not reflected in Wei et al.'s paper. There are also differences in specific analysis methods. The genome data of roses was published in 2017, and with the development of bioinformatics technology, gene structure annotations have been manually corrected. In this article, we employed the GSAman tool (https://gitee.com/cjchen/igv-srna) to rectify the gene structure of TPS and TPP gene family members prior to conducting our analysis. It is believed that this is not only the first accurate identification and analysis of the rose TPP gene family, but also the enrichment and improvement of the rose TPS family. In addition, we analyzed the expression patterns of identified TPS and TPP gene families using both transcriptome data and qPCR expression level data. In terms of specific expression patterns, we simultaneously investigated tissue-specific expression and light-induced expression after environmental treatment, resulting in a more comprehensive and detailed dataset. In addition, when analyzing the promoter regions, we pay more attention to the specific types of cis acting elements, while Wei focuses on the functions. A recently published article confirms our accurate prediction of the interaction between the TPP family and myb transcription factors [38]. We will also update in the paper.

Finally, based on previous observations and measurements of flowering phenotypes and indicators in roses under different light environments, our study found that trehalose may also be involved in the flowering response of roses under low light conditions, which has never been reported in roses, especially since the combined involvement of light and sugar signals in the flowering process of roses has not yet been revealed. Unlike the TPS gene family reported separately by Wei et al., we focus more on the key roles of TPS, TPP, and Tre in the flowering of roses under light environments.

We will also present our method, experimental design, and results in detail in the paper, and compare and discuss them with existing related studies to highlight the uniqueness and contribution of our research. Thank you again for the guidance and feedback from the reviewer.

[19] Raza, A.; Bhardwaj, S.; Atikur Rahman, M.; García-Caparrós, P.; Habib, M.; Saeed, F.; Charagh, S.; Foyer, C.H.; Siddique, K.H.M.; Varshney, R.K. Trehalose: A sugar molecule involved in temperature stress management in plants. The Crop Journal 2023, doi:https://doi.org/10.1016/j.cj.2023.09.010.

[27] Gao, Y.H.; Yang, X.Y.; Yang, X.; Zhao, T.Y.; An, X.M.; Chen, Z. Characterization and expression pattern of the trehalose-6-phosphate synthase and trehalose-6-phosphate phosphatase gene families in Populus. International Journal of Biological Macromolecules 2021, 187, 9-23, doi:10.1016/j.ijbiomac.2021.07.096.

[38] Yu, H.; Teng, Z.; Liu, B.; Lv, J.; Chen, Y.; Qin, Z.; Peng, Y.; Meng, S.; He, Y.; Duan, M.; et al. Transcription factor OsMYB30 increases trehalose content to inhibit alpha-amylase and seed germination at low temperature. Plant physiology 2023, doi:10.1093/plphys/kiad650.

Comments 3:

  I think it would be better to emphasize the novelty of this paper a little more.

Response 3:

Thank you very much for your attention and suggestions on our research. We attach great importance to your feedback on emphasizing the novelty of the paper and believe that it is of great significance to our research.

Firstly, we will clearly articulate the innovative points and contributions of this study in the revised version. We will provide a more detailed description of our methods and results, as well as their relevance to existing research, in the abstract and introduction sections. This will help readers better understand the value and novelty of our research.

In addition, we will discuss in detail the similarities and differences between this study and other related studies in the Materials Methods and Discussion section, as well as our improvements in methods or results. This will help highlight the uniqueness and innovation of our research in existing research.

The specific modifications are as follows:

line: 21-39 (Abstract)

Abstract: Trehalose, trehalose-6-phosphate synthase (TPS), and trehalose-6-phosphatase (TPP) have been reported to play important roles in plant abiotic stress and growth development. However, their functions in the flowering process of Rosa hybrida have not been characterized. In this study we found that under short photoperiod or weak light intensity, the content of trehalose in the shoot apical meristem of Rosa hybrida cv 'Carola' significantly decreased, leading to delayed flowering time A total of 9 RhTPSs and 7 RhTPPs genes were identified in the genome. Cis-element analysis suggested that RhTPS and RhTPP genes were involved in plant hormones and environ-mental stress response. Transcriptome data analysis reveals significant differences in the expression levels of RhTPSs and RhTPPs family genes in different tissues, and indicates that RhTPPF and RhTPPJ are potential key genes involved in rose flower bud development under different light environments. The results of quantitative real-time reverse transcription (qRT-PCR) further in-dicate that under short photoperiod and weak light intensity all RhTPP members were significantly down-regulated. Additionally, RhTPS1a, RhTPS10, and RhTPS11 were up-regulated under short photoperiod and showed a negative correlation with flowering time and trehalose content decrease. Under weak light intensity, RhTPS11 was up-regulated and negatively regulated flowering, while RhTPS5, RhTPS6, RhTPS7b, RhTPS9, and RhTPS10 were down-regulated and positively regulated flowering. This work lays the foundation for revealing the functions of RhTPS and RhTPP gene families in the regulation of rose trehalose.

line: 107-122 (introduction)

Rosa hybrida is currently the largest cut flower in terms of global planting area and sales volume [18]. The TPS and TPP gene families have been reported to be of great significance in many kinds of plants response to stress and regulation of growth and development [19,20]. With the update of gene structure annotation technology(https://gitee.com/CJchen/IGV-sRNA), and so rose TPS genes need be reidentified and analyzed. Furthermore, there have been no studies to evaluate the evolution and expression profiles of the TPP gene family in Rosa hybrida. In this study, the member of TPS and TPP family were identified in rose, and their structures, phylogenetic relationships,cis-acting elements were thoroughly analyzed. The expression patterns of TPS and TPP in different tissues of roses were analyzed using transcriptome data. Additionally, the crucial role of RhTPS and RhTPP in flower formation in response to various photoperiods and light intensities were investigated by transcriptome data and qRT-PCR. Finally, we conducted a correlation analysis between flowering time, trehalose content, and RhTPSs/TPPs gene expression to identify key functional genes. The findings from this study serve as a valuable biological foundation for gaining a deeper understanding of the functionalities of the RhTPS and RhTPP gene families in rose.

  1. Raymond, O.; Gouzy, J.; Just, J.; Badouin, H.; Verdenaud, M.; Lemainque, A.; Vergne, P.; Moja, S.; Choisne, N.; Pont, C.; et al. The Rosa genome provides new insights into the domestication of modern roses. Nature Genetics 2018, 50, 772-777, doi:10.1038/s41588-018-0110-3.
  2. Raza, A.; Bhardwaj, S.; Atikur Rahman, M.; García-Caparrós, P.; Habib, M.; Saeed, F.; Charagh, S.; Foyer, C.H.; Siddique, K.H.M.; Varshney, R.K. Trehalose: A sugar molecule involved in temperature stress management in plants. The Crop Journal 2023, doi:https://doi.org/10.1016/j.cj.2023.09.010.
  3. Fichtner, F.; Lunn, J.E. The Role of Trehalose 6-Phosphate (Tre6P) in Plant Metabolism and Development. Annu Rev Plant Biol 2021, 72, 737-760, doi:10.1146/annurev-arplant-050718-095929.

line: 428-431 (Discussion)

Transcriptional activation levels are coordinated by upstream cis-acting elements, which are crucial for plant responses to environmental conditions. The RhTPS and RhTPP promoters contain some common motifs and numerous repetitive regions. All RhTPS and RhTPP promoters contain MYB elements (Fig. 5), which are important for plant development and stress response [36,37]. Inducing OsMYB30 through low-temperature signaling in rice, which activates the expression of OsTPP1 gene, leads to the accumulation of fucose in seeds and α- Inhibition of amylase activity, thereby inhibiting seed germination at low temperatures [38].

  1. Liu, J.; Osbourn, A.; Ma, P. MYB Transcription Factors as Regulators of Phenylpropanoid Metabolism in Plants. Molecular plant 2015, 8, 689-708, doi:10.1016/j.molp.2015.03.012.
  2. Wei, X.-R.; Ling, W.; Ma, Y.-W.; Du, J.-L.; Cao, F.-X.; Chen, H.-X.; Chen, J.-R.; Li, Y.-F. Genome-Wide Analysis of the Trehalose-6-Phosphate Synthase Gene Family in Rose (<i>Ros</i><i>a chinensis</i>) and Differential Expression under Heat Stress. Horticulturae 2022, 8, doi:10.3390/horticulturae8050429.
  3. Yu, H.; Teng, Z.; Liu, B.; Lv, J.; Chen, Y.; Qin, Z.; Peng, Y.; Meng, S.; He, Y.; Duan, M.; et al. Transcription factor OsMYB30 increases trehalose content to inhibit alpha-amylase and seed germination at low temperature. Plant physiology 2023, doi:10.1093/plphys/kiad650.

line: 444-460 (Discussion)

Added discussion on qPCR data.

3.4 Expression patterns of RhTPSs and RhTPPs under different light treatment

We further investigated the expression patterns of RhTPS and RhTPP genes under different light treatments. As shown in Figure 6c and 6d, many RhTPP genes and some RhTPS genes may be related to various aspects of low light stress. In addition, most of these genes may respond to both short photoperiod and low light intensity treatments. For example, RhTPS9,10 and RhTPPs (Figure 6c and 6d). In Arabidopsis, TPS9 has the function of regulating flower transformation in leaf vascular tissues, long day light cycle stimulation leads to increased chromatin accessibility and increased expression of TPS9 [42]. In roses, reducing light exposure time appropriately (10 hours of light) and decreasing light intensity (80% of light intensity) can promote the expression of RhTPS9. Therefore, the mode of action of TPS9 may be inconsistent in roses and Arabidopsis. We speculate that RhTPS9,10 and RhTPPs may play a crucial role in the flowering of roses in response to different light environments. The expression of RhTPPs in response to light environment is relatively consistent (Figure 6d), indicating that RhTPPs may have consistent functions. In contrast, all 13 MdTPS genes in apples responded to exogenous sucrose application, indicating that TPS may play a consistent role in apple response to sucrose treatment [43]. This also reflects the differential expression of TPS and TPP in different species.

  1. Tian, H.; Li, Y.; Wang, C.; Xu, X.; Zhang, Y.; Zeb, Q.; Zicola, J.; Fu, Y.; Turck, F.; Li, L.; et al. Photoperiod-responsive changes in chromatin accessibility in phloem companion and epidermis cells of Arabidopsis leaves. The Plant Cell 2021, 33, 475-491, doi:10.1093/plcell/koaa043 %J The Plant Cell.
  2. Du, L.; Qi, S.; Ma, J.; Xing, L.; Fan, S.; Zhang, S.; Li, Y.; Shen, Y.; Zhang, D.; Han, M. Identification of TPS family members in apple (Malus x domestica Borkh.) and the effect of sucrose sprays on TPS expression and floral induction. Plant physiology and biochemistry: PPB 2017, 120, 10-23, doi:10.1016/j.plaphy.2017.09.015.

We would like to thank the referee again for taking the time to review our manuscript.

Reviewer 2 Report

Comments and Suggestions for Authors

The last paragraph of the introduction should contain the aim of the study. Instead, the writers gave the need for work followed by the conclusion. There is no aim, objective, or scope of the study.

No detail about the statistical analysis in M&M

The source of the plant material is not given. 

Author Response

For research article

Response to Reviewer 2 Comments

1. Summary

Dear Reviewer, thank you very much for taking the time to review our manuscript. We greatly appreciate your professional insights and valuable suggestions. Below, we have provided detailed responses to your comments and made corresponding revisions and corrections in the resubmitted files. We sincerely hope that these modifications meet your expectations and enhance the quality of our research. For clarity, we will use red text to show our revisions.

2. Point-by-point response to Comments and Suggestions for Authors

Comments 1:

The last paragraph of the introduction should contain the aim of the study. Instead, the writers gave the need for work followed by the conclusion. There is no aim, objective, or scope of the study.

Response 1:

Thank you very much for your question and helping us improve the quality of our manuscript. As suggested, we have rewritten the last paragraph of the introduction to emphasize our research objectives, aims, and scope, enabling readers to accurately understand our research intent. This change can be found in the revised version of the article (line84-99). The specific modifications are as follows:

Rosa hybrida is currently the largest cut flower in terms of global planting area and sales volume [18]. The TPS and TPP gene families have been reported to be of great significance in many kinds of plants response to stress and regulation of growth and development [19,20]. With the update of gene structure annotation technology(https://gitee.com/CJchen/IGV-sRNA), and so rose TPS genes need be reidentified and analyzed. Furthermore, there have been no studies to evaluate the evolution and expression profiles of the TPP gene family in Rosa hybrida. In this study, the member of TPS and TPP family were identified in rose, and their structures, phylogenetic relationships,cis-acting elements were thoroughly analyzed. The expression patterns of TPS and TPP in different tissues of roses were analyzed using transcriptome data. Additionally, the crucial role of RhTPS and RhTPP in flower formation in response to various photoperiods and light intensities were investigated by transcriptome data and qRT-PCR. Finally, we conducted a correlation analysis between flowering time, trehalose content, and RhTPSs/TPPs gene expression to identify key functional genes. The findings from this study serve as a valuable biological foundation for gaining a deeper understanding of the functionalities of the RhTPS and RhTPP gene families in rose.

18. Raymond, O.; Gouzy, J.; Just, J.; Badouin, H.; Verdenaud, M.; Lemainque, A.; Vergne, P.; Moja, S.; Choisne, N.; Pont, C.; et al. The Rosa genome provides new insights into the domestication of modern roses. Nature Genetics 2018, 50, 772-777, doi:10.1038/s41588-018-0110-3.

19.   Raza, A.; Bhardwaj, S.; Atikur Rahman, M.; García-Caparrós, P.; Habib, M.; Saeed, F.; Charagh, S.; Foyer, C.H.; Siddique, K.H.M.; Varshney, R.K. Trehalose: A sugar molecule involved in temperature stress management in plants. The Crop Journal 2023, doi:https://doi.org/10.1016/j.cj.2023.09.010.

20.   Fichtner, F.; Lunn, J.E. The Role of Trehalose 6-Phosphate (Tre6P) in Plant Metabolism and Development. Annu Rev Plant Biol 2021, 72, 737-760, doi:10.1146/annurev-arplant-050718-095929.

Comments 2:

No detail about the statistical analysis in M&M

Response 2:

We apologize for not providing detailed information on statistical analysis in the "Materials and Methods" section. To address this issue, we will make necessary revisions to the "Materials and Methods" section to provide more comprehensive details on statistical analysis. We will explicitly state the statistical methods employed, the analytical steps taken, and the relevant software used.

4.8 Data analysis

The data were analyzed using GraphPad Prism 9.0 (GraphPad Software, Boston, Massachusetts USA, www.graphpad.com) in order to compare the differences between treatments. The figures display the mean ± standard deviation (SD) of biological triplicates, unless otherwise indicated. One-way analysis of variance was employed to determine statistical significance.

Correlation analysis was adopted using Pearson’s correlation. The package ggcorrplot was used for visualizations. p-Values less than 0.05 (p < 0.05) were displayed.

Comments 3:

The source of the plant material is not given.

Response 3:

Thank you for your comment. Indeed, the source of plant materials is crucial information in research to ensure the reproducibility and verifiability of experiments. We will clarify the source of materials and the mode of reproduction in the materials and methods.

4.1. Plant materials and treatments

In this study, Rosa hybrida cv ‘Carola’ was obtained from the Key Laboratory of Cold Region Landscape and Application, Northeast Agricultural University, Harbin, China (45.6811°N,126.6250°E). Using cuttage for propagation expansion.

We would like to thank the referee again for taking the time to review our manuscript.

Round 2

Reviewer 2 Report

Comments and Suggestions for Authors

All revisions are done appropriately